# Microplastic Distribution through the Salinity Gradient in a Stratified Estuary

**Marija Parać** [ID], **Vlado Cuculić** [ID], **Nuša Cukrov, Sunčana Geček, Marin Lovrić and Neven Cukrov** *[ID]

Ruđer Bošković Institute, Division for Marine and Environmental Research, Bijenička cesta 54, 10000 Zagreb, Croatia

\* Correspondence: ncukrov@irb.hr

**Abstract:** Despite the extensive and rapidly growing literature on microplastics in oceans and coastal seas, little information exists on microplastic distribution through the salinity gradient. This study is the first one to evaluate microplastic distribution through the salinity gradient of a highly stratified estuary. A total of 910 microplastic particles were collected from 12 different sampling events in the Krka River estuary, Croatia. The number of detected particles ranged from 389 in the surface layer to 63 in the deepest marine layer. The highest plastic abundance was found in the surface layer (3.68 particles/m$^3$) and the lowest in the marine layer (0.13 particles/m$^3$). The measured values of the cross-sectional area indicated an ellipsoidal cross-sectional shape of the particles. It was also found that the majority of microplastic particles belonged to the small microplastic class (<1 mm). The Nile Red (NR) staining method was used to visualize fluorescent microplastic particles, while quantification was performed using ImageJ/Fiji software. The strong salinity stratification in the studied estuary did not alter the usual distribution of microplastic particles in the water column, and there was no significant accumulation on the halocline.

**Keywords:** microplastics; salinity gradient; Krka River estuary; Nile Red (NR); fluorescence





## 1. Introduction

Microplastics are plastic particles less than 5 mm in diameter. Many authors have given different classifications for microplastics with numerous size ranges, but the lower limit has not yet been agreed upon. However, microplastics have been recognized as a ubiquitous anthropogenic contaminant in aquatic ecosystems. Owing to their durability, they remain in the environment for a long period of time, during which they can be translocated by winds or currents, changing their distribution through the water column [1]. Furthermore, microplastics are prone to adsorption of toxic pollutants (e.g., heavy metals, persistent organic pollutants (POPs), polycyclic aromatic hydrocarbons (PAHs), polychlorinated biphenyls (PBCs), phthalates) [2] and biofouling [3] that leads to changes in initial density and affecting buoyancy. Most research has focused on assessing microplastic pollution in the surface (SL) or subsurface water layers (SBL) of the marine environment [4], while other water layers at various depths remain unexplored. There is evidence that microplastics can be transported along the water column by vertical mixing despite their density relative to seawater [5]. Environmental factors such as storms, wind-driven mixing events, resuspension, and attachment appear to play an important role in the vertical distribution of microplastics [1]. To avoid under- or overestimating the abundance of microplastics, it is important to evaluate samples from different depths in the water column. The occurrence of microplastics is even more evident in urbanized estuaries with limited water exchange and high anthropogenic pressures originating from tourism, wastewater emissions, boat leisure activities, fishing, mariculture, runoff waters, and harbors [6].

To assess the occurrence and vertical transport of microplastics in estuaries with limited water exchange, the Krka River estuary was chosen as the study site. It is permanently

vertically stratified with three separate layers in a vertical profile: the upper brackish layer (river water layer), the freshwater-seawater interface (FSI; middle layer), and the bottom seawater layer. The halocline with a strong vertical salinity gradient usually varies by shape due to location, weather (wind, precipitation), and hydrological (Krka River flow) conditions [7]. It is believed to have a major influence on the vertical distribution of plastic particles in the water column. In addition, salt-wedge estuaries have turbulent mixing events that can drastically affect the behavior of water masses and suspended particles, followed by strong stratification periods [8]. The distribution of microplastics in this area has not been investigated yet, but other studies emphasize the influence of salinity gradient on the vertical distribution of microplastics in the water column [1,8,9].

This research focuses on the morphological characterization of microplastics ranging in size from 30 μm to 5 mm and their differential vertical distribution in the water column. In doing so, we tested the hypothesis that the strong salinity stratification in the Krka River estuary may alter the usual distribution of microplastics in the water column, so that most of them accumulate on the halocline layer. This study provides a notable insight into the vertical distribution and accumulation of microplastics in the water column in the Krka River estuary.

## 2. Materials and Methods

All methods and protocols used were adapted from the National Oceanic and Atmospheric Administration (NOAA) [10] and from the multinational BASEMAN project funded under the EU Joint Program Initiative (JPI) Ocean [11].

### 2.1. Study Area

The Krka River Estuary is positioned in the central part of the eastern Adriatic coast, from the Skradinski Buk waterfall (43°48′ N; 15°55′ E) in the north to the St. Nicholas Fortress (43°43′ N; 15°51′ E) in the south. The estuary is 23.5 km long, has a complex morphology and varying depth from 5 m (below Skradinski Buk waterfall) to 43 m (at the mouth) [12,13]. There is permanent vertical stratification between the bottom seawater and the surface fresh or brackish water due to the very low tidal amplitude (20–30 cm) that occurs in the Mediterranean Sea [14] as well as its sheltered location [15]. Three separate layers are easily distinguished: the upper brackish layer (river water layer), the freshwater-seawater interface (FSI; middle layer), and the bottom seawater layer [7]. The upper brackish water current flows toward the sea, while the deeper saltwater current flows in the opposite direction. A strong halocline defines the boundary layer, which differs in thickness and depth depending on freshwater inflow and wind [15]. Moreover, the halocline prevents the exchange between the brackish water at the surface and the bottom seawater, acting like a barrier [16]. According to [17], the estimated exchange time for freshwater in the Krka River estuary is up to 20 days during winter and up to 80 days during summer, while the exchange time for seawater is 50 to 100 days during winter and up to 250 days in July and August. The freshwater flow from the Krka River varies monthly and seasonally from 20 to 170 $m^3/s$, sometimes from 5 $m^3/s$ to 440 $m^3/s$ [15], with an average of 55 $m^3/s$ [13].

The city of Šibenik, located in lower part of the Krka River estuary, has a population of around 31,000 inhabitants [18]. However, due to tourism, especially nautical tourism [19], these numbers multiply during the summer season. The port/harbor of the city includes the transportation of phosphate ore [12], wood, and aluminum. There are also several mussel (*Mytilus galloprovincialis*) farm plants [20].

### 2.2. Quality Assurance and Quality Control (QA/QC)

Due to their ubiquitous nature, microplastic particles could easily contaminate the samples we work with, which could lead to an overestimation of the abundance of microplastics in the sample. Given this, it is critical to avoid any cross-contamination by implementing several contamination minimization procedures, as we have, such as wear-

ing a clean 100% cotton laboratory coat and nitrile gloves; avoiding wearing synthetic clothing, even under the coat, especially fleece; closing doors and windows to minimize air movement in the laboratory; cleaning all equipment and working stations with 70% ethanol and rinsing three times with Milli-Q® water; washing glassware with 10% $HNO_3$ and rinsing three times with Milli-Q® water; covering all equipment and samples with aluminum foil; inspecting all Petri dishes, filters, and forceps under a stereomicroscope; using non-plastic material (steel, glass, aluminum); working in a fume hood; pre-filtering all working solutions and reagents with LLG Syringe Filters SPHEROS, PTFE (pore size: 0.22 µm); and carrying out field, laboratory and procedural blanks.

*2.3. Microplastics Sampling*

Samples were collected in the lower part of the Krka River estuary (Figure 1) in June 2022 on a clear day, wind speed 3.0 m/s (direction: SW), air temperature 30 °C. Water temperature was 23 °C, Beaufort scale 2 (light breeze), Douglas scale 1 (calm (rippled)).

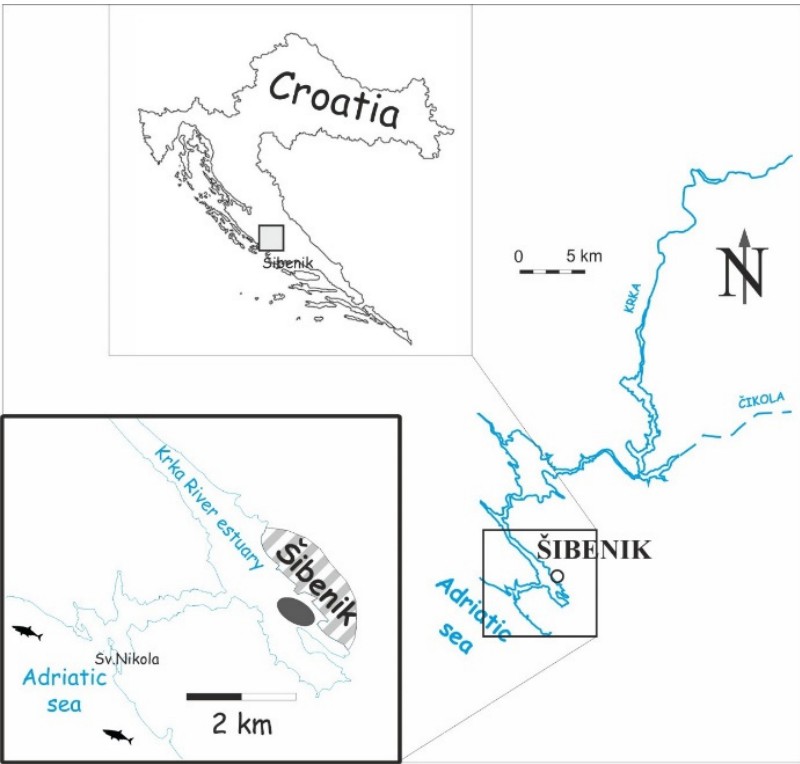

**Figure 1.** Map of the study site; dark ellipse indicates sampling area.

Salinity was measured with a refractometer at the surface (15) and at the halocline (31). The average vessel speed was 3 km/h and the duration time for towing was 5 min. Volume-reduced samples of the SL were obtained by submerging only half of the "Net for Microplastic Sampling" (Hydro-Bios, Apparatebau GmbH, Altenholz, Germany; mesh size: 300 µm; net aperture: 0.28 m² [width 70 cm, height 40 cm, length 260 cm]). SBL samples were obtained by fully immersing the "Net for Microplastic Sampling" which sampled the first 40 cm of the water column. For sampling at the freshwater-seawater interface (halocline layer (HL), depth 1.5 m), a scuba diver controlled the net. For the marine layer (ML) (2 m), the net was towed without buoys. A total of 12 samples were collected, 3 replicates for each layer in the water column. An average of 166.95 m³ of seawater was collected at each sampling point using a "Net for Microplastic Sampling". The volume of water that flowed through the net was calculated using a flow meter (Mechanical Flow Meter, Hydro-Bios, Apparatebau GmbH, Germany) attached to the net rim, according to the given manual by the manufacturer. After towing, the net externally rinsed with Milli-Q® water from

a pressure container to avoid contamination of the samples. All sampled microplastic particles were collected in a glass jar with a lid from the cod end. Samples were refrigerated at +4 °C until further processing in the laboratory.

### 2.4. Sieving

Obtained samples from glass jars were wet sieved through a 250 μm sieve, thoroughly rinsed with Milli-Q® water to collect all microplastic particles, and transferred to a new clean and labeled glass jar. The samples were divided into 2 groups: >250 μm and <250 μm.

### 2.5. Organic Matter Removal

For organic matter removal, Fenton's reagent (a mixture of 0.05 M Fe (II) sulfate (7.5 g of $FeSO_4 \cdot 7H_2O$ (from Gram-Mol d.o.o.) in 500 mL of Milli-Q® water and 3 mL of concentrated sulfuric acid (from Acros Organics) with a 30% $H_2O_2$ solution (from Gram-Mol d.o.o.)) was used (pH = 2.8). To each beaker, 40 mL of Fenton reagent was added and heated on a "hot plate" at 75 °C for 2 h. Some samples contained larger amounts of organic matter, so a second addition of 20 mL of $H_2O_2$ solution was required for complete removal.

### 2.6. Density Separation

A saturated salt solution (1.2 $g/cm^3$) was prepared by dissolving 360 g of NaCl in 1000 mL of Milli-Q® water and placing it on a heated magnetic stirrer for 30 min to fully dissolve. The solution was filtered through glass microfiber filters (LGG Labware; pore size 1.6 μm; filter diameter Ø 47 mm) mounted on a filtration system (MF31, Rocker Scientific, Kaohsiung City, Taiwan) connected to a vacuum pump (Büchi® V-500, Merck, Ciudad Autónoma de Buenos Aires Argentina); 100 mL of saline solution and the sample were poured into a clean beaker, placed on a magnetic stirrer for 2 min, and sonicated for 15 min (Sonorex Super RK 255 H, Bandelin, Ciudad Autónoma de Buenos Aires, Argentina). The solution was allowed to sediment for at least 2 h while covered with aluminum foil to prevent airborne contamination. The supernatant containing microplastics was then transferred to a clean beaker. The walls of a density separator were thoroughly rinsed with Milli-Q® water to transfer all remaining microplastic particles. This step was repeated 2 times to increase the recovery rate of plastic particles.

### 2.7. Filtration

All samples and solutions were filtered through glass microfiber filters (LGG Labware; pore size 1.6 μm; filter diameter Ø 47 mm) placed on a filtration system (MF31, Rocker Scientific) connected to a vacuum pump (Büchi® V-500). The funnel walls were thoroughly rinsed with Milli-Q® water to transfer all remaining microplastic particles to the filters.

### 2.8. Nile Red (NR) Staining

This protocol was developed by combining several research and review papers [21–27] and optimized for our laboratory conditions.

Nile Red (NR) is a hydrophobic, lipophilic, solvatochromic, and photochemically stable dye often used in microplastic research since it fluoresces strongly in hydrophobic solvents. This staining dye ideally distinguishes plastic material from mineral or organic matter in the sample, which aids in the microplastic visualization and identification. The color of NR fluorescence varies from deep red to golden yellow [26]. However, this method is not a replacement for analytical methods such as FTIR or Raman spectroscopy, as it tends to give false-positive results and is not able to determine the chemical composition of the samples (i.e., polymer type).

NR (Cas-No: 7385-67-3, technical grade, Sigma-Aldrich, Merck, Ciudad Autónoma de Buenos Aires Argentina) stock solution was prepared at 1 mg/mL in methanol (Cas-No: 67-56-1; p.a., Gram-mol d.o.o.) and filtered into a clean glass volumetric flask (10 mL) wrapped in aluminum foil using a non-sterile 0.22 μm polytetrafluoroethylene (PTFE) syringe filter (Cas-No: 6.272 818, LLG Syringe Filters SPHEROS, PTFE; Ø 25 mm). The

stock solution was stored in the freezer ($-4$ °C) until further use. 0.1 mL of the Nile Red stock solution (1 mg/mL) was used per 100 mL of methanol, giving a final concentration of 1 µg/mL for microplastic staining, respectively.

All filters in glass Petri dishes were analyzed using a Nikon SMZ745T stereomicroscope fitted with the MZX-B-LED light source (Blue; excitation wavelength: 420–490 nm; emission wavelength: >495 nm) (Guangzhou Micro-shot Technology Co., Ltd., Guangzhou, China) and a Bresser MikroCam PRO HDMI 5 MP digital microscope camera head.

The microplastic-containing filters were stained by adding 1 mL of NR working solution (1 µg/mL) directly to the filters positioned on the filtration system and waiting 5 min. The filters were then thoroughly rinsed with Milli-Q® water to wash away unabsorbed dye particles that could interfere with fluorescence. The filters were then incubated in the dark at room temperature for 30 min. The filters in the Petri dishes were covered with aluminum foil and placed in a fume hood to dry overnight. After drying, the filters were ready for quantification of the fluorescent microplastics. An appropriate drying step was necessary because the presence of water decreases fluorescence.

### 2.9. ImageJ/Fiji Image Processing

Visual inspection of each putative microplastic particle was conducted using a stereomicroscope (Nikon SMZ745T, Tokyo, Japan) equipped with Bresser MikroCam PRO HDMI 5 MP. For the acquisition of images, MikroCamLabII version 4.7.15283 (Bresser, GmbH, Rhede, Germany) software was used. The microscope was covered with a cardboard box and black cloth to prevent any light from contaminating the sample. To avoid loss of fluorescence, all samples were processed as soon as possible and kept in a dark room with constant temperature. First, all filters were checked for autofluorescence before the dye was applied because sometimes some materials or organisms fluoresce under certain wavelengths. The same particle was photographed under bright light and blue LED light. The color emission varied from deep red to golden yellow, depending on the surface hydrophobicity of the plastic particle, which can be altered due to weathering or surface contamination by the environment [21,28]. Same settings for photo acquisition were set for each sample, including automatic exposure time, white balance (capturing a blank filter with no particles in bright field), dark field correction (to allow subtraction of noise caused by the camera), and the same magnification (6.7×). All images were saved in TIFF format to avoid loss compression and to preserve original raw image data. Image processing was done by using ImageJ/Fiji software [29] based on the script developed by [24,27] with some modifications. Images of the stained filters were measured to determine the Maximum Feret diameter (mm) and cross-sectional area (mm$^2$). The same scale was used for all images (Analyze > Set Scale > 112 px/mm). A specific macro was recorded in ImageJ/Fiji software for automatic counting of particles and later used for batch processing of all images to speed up quantification. Automatic quantification allows for objectivity, speed, reproducibility, comparability, and high throughput. However, there are some pitfalls, e.g., overlapping particles are often counted as a single particle, counting noise by the camera as microplastics if the threshold is set too low, the over- or underestimation of particles, and the lack of detection of color, polymer type, or exact shape. All images were opened as greyscale 8-bit images with separate channels (RGB) using the Bio-Formats plugin. The green channel was considered for quantification in blue light. The auto threshold was set using the RenyiEntropy algorithm to avoid user bias during choosing threshold cut off values. The minimum size was set to 30 µm due to the limited camera resolution. It can be improved with more powerful lenses or microscopes. The results were exported as xlsx files.

### 2.10. Data Analysis

Statistical analyses and graphical representation of results were performed in the R statistical environment (R version 3.6.3, R Studio 1.2.5033). The *base* [30] and *pastecs* [31] packages were used for descriptive statistics and basic homoscedasticity (Levene test),

normality (Shapiro–Wilks, QQ-plot), and extreme value tests. The *ggplot2* package [32] was used for graphical display. Robust ANOVA based on trimmed means (20% of trimming level [33]; *WRS2* package in R) was used to test the difference in particle size between water layers instead of classical ANOVA to overcome the problems associated with deviations from homoscedasticity and to reduce the influence of outliers. Particle size data were log-transformed prior to omnibus testing. Post hoc testing was also performed in the robust *WRS2* environment, where *p*-values were adjusted for multiple testing using the Benjamini-Hochberg (BH) method. The *multcompView* package [34] was used to convert the vector of *p*-values to a character-based display in which common characters denote levels or groups that are not significantly different.

## 3. Results

In this study, a total of 910 microplastic particles were collected from 12 sampling events. Microplastics were found in all samples. As shown in Table 1, the number of detected particles ranged from 389 in SL, to 372 in SBL, 86 in HL, and 63 in ML. The total sampled volumes varied from 108.07 m$^3$ to 980.00 m$^3$. The highest plastic abundance was found in SL (3.68 particles/m$^3$), followed by the SBL (0.38 particles/m$^3$), then in HL (0.19 particles/m$^3$), and the lowest in ML (0.13 particles/m$^3$) (Figure 2A).

**Table 1.** Overview of microplastic particles distribution at different layers across the salinity gradient. Mean values were calculated in respect to 12 sampling events (three in each layer).

| Layer | Total Volume (m$^3$) | Total Number of Particles | Total Plastic Cross-Sectional Area (mm$^2$) | Mean Abundance of Plastics (# Particles/m$^3$) | Mean Area of Plastics per Water Volume (m$^2$/m$^3$) |
|---|---|---|---|---|---|
| SL | 108.07 | 389 | 113.73 | 3.68 | 1.054 |
| SBL | 980.00 | 372 | 185.23 | 0.38 | 0.191 |
| HL | 445.76 | 86 | 26.11 | 0.19 | 0.058 |
| ML | 469.56 | 63 | 11.14 | 0.13 | 0.023 |

SL—surface layer; SBL—subsurface layer; HL—halocline; ML—marine layer.

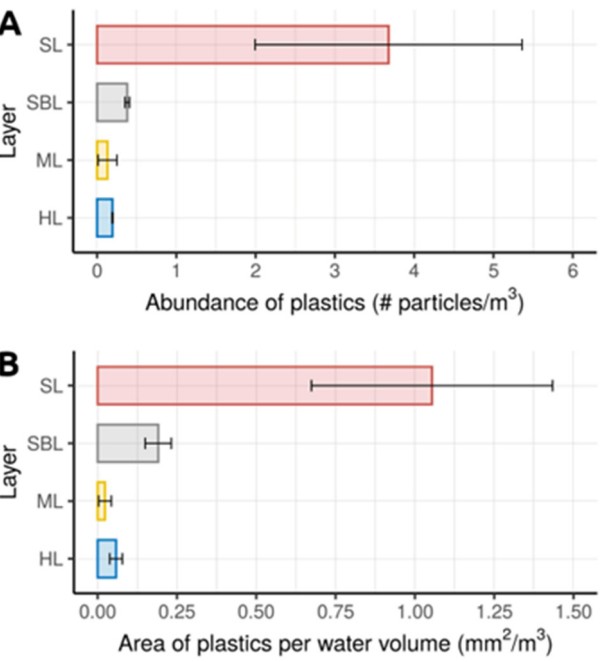

**Figure 2.** Presence of plastics in the water column expressed as (**A**) abundance of particles and (**B**) observed area of particles in the water volume. Error bars represent the mean standard error of sampling events.

The total observed area of collected particles was 336.21 mm$^2$ overall, with the maximum of 185.23 mm$^2$ found for SBL and the minimum of 11.14 mm$^2$ for ML. Mean observed area of plastics in water volume showed the characteristic exponential decrease with depth in a manner similar to the vertical abundance of particles (Figure 2A,B): the observed area per water volume was the largest in the SL (1.054 mm$^2$/m$^3$), followed by the SBL (0.191 mm$^2$/m$^3$), then (0.058 mm$^2$/m$^3$) in HL, and the smallest in ML (0.023 mm$^2$/m$^3$).

Two size measures were obtained for each particle: maximum Feret diameter (mm) and cross-sectional area (mm$^2$), the latter being more precise because it is based on two-dimensional information, as opposed to one-dimensional diameter. Most of the values of the measured particle area clustered around the left tail of the distribution (<0.4 mm$^2$), while the right tail of the distribution was longer representing smaller number of considerably larger particles with areas up to 20 mm$^2$ (Figure 3A, Table 2). This resulted in a significant deviation of the median values from the mean values under the influence of outliers and positive skewness (Table 2). Log-transformation normalized the distributions and attenuated the influence of outliers, revealing a significant difference in particle size between layers by robust ANOVA (F (3, 131.52) = 5.3661, *p* = 0.00162, effect size ξ = 0.24, CI$_{95}$(ξ) = [0.07; 0.36], n$_{obs}$ = 910) (Figure 3C). Benjamini–Hochberg robust post hoc tests revealed differences in size between the upper layers (SL, SBL) and lower layers (HL, ML): SL vs. HL (ψ̂ = −0.168, *p* = 0.014), SL vs. ML (ψ̂ = −0.13, *p* = 0.042), SBL vs. HL (ψ̂ = −0.203, *p* = 0.007), and SBL vs. ML (ψ̂ = −0.166, *p* = 0.016). However, there was no significant difference in the upper layer between SL and SBL (ψ̂ = 0.036, *p* = 0.504) or in the lower layer between HL and ML (ψ̂ = 0.037, *p* = 0.605).

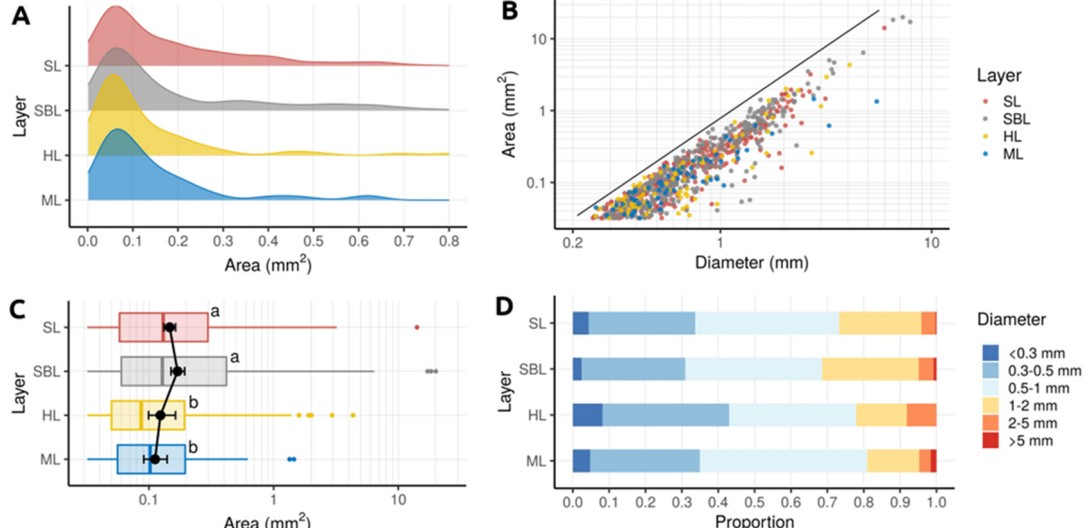

**Figure 3.** (**A**) Size distribution (area) of particles in each layer (**B**) Boxplot representation of size (area) in logarithmic scale. Significant differences between log-transformed groups are indicated with different letters, and error bars represent 95% confidence intervals (CIs) around the mean. (**C**) Two measures of particle size: diameter and area. The line represents the theoretical relationship between diameter and area when the cross-section of the particle is circular (area = π/4 × diameter$^2$). (**D**) Distribution of particle size (diameter) among six size classes.

**Table 2.** Descriptive statistics of the size of particles in each layer. Size is expressed as observed cross-sectional area of particles. N represents the number of particles sampled in each layer, i.e., sample size.

| Layer | N | Min (mm$^2$) | Max (mm$^2$) | Sum (mm$^2$) | Median (mm$^2$) | Mean (mm$^2$) | Std (mm$^2$) |
|-------|-----|-------|--------|---------|-------|-------|-------|
| SL | 389 | 0.030 | 14.133 | 113.730 | 0.128 | 0.292 | 0.791 |
| SBL | 372 | 0.030 | 20.002 | 185.232 | 0.123 | 0.498 | 1.753 |
| HL | 86 | 0.031 | 4.335 | 26.114 | 0.080 | 0.304 | 0.656 |
| ML | 63 | 0.031 | 1.455 | 11.142 | 0.099 | 0.177 | 0.258 |

Although cross-sectional area is more precise, diameter is the most commonly cited measure of particle size in the literature and in the classification of microplastics. Therefore, the degree of association between diameter and area was determined by estimating the correlation coefficient. The Pearson's correlation coefficient between squared diameter and area was exceptionally high in the upper layers SL (r = 0.96, CI$_{95}$ = [0.95; 0.97]) and SLB (r = 0.98, CI$_{95}$ = [0.97; 0.98]), while the values in the lower layers were slightly lower: HL (r = 0.91, CI$_{95}$ = [0.86; 0.94]) and ML (r = 0.79 CI$_{95}$ = [0.68; 0.87]). Overall, the measured values of area were on average 58–62% smaller than the theoretical area of the circle whose radius would be half of the measured diameter (Figure 3B), indicating (i) an ellipsoid cross-sectional shape of the particles and (ii) suggesting that we should be cautious in interpreting particle size when using area and diameter interchangeably. Finally, it was found that the majority of particles (98.4–100%) belonged to the microplastic class (diameter < 5 mm), with the small microplastic class (diameter < 1 mm) being more dominant (68.5–80.9%) than the large microplastic class (1 mm < diameter < 5 mm) (Figure 3D).

## 4. Discussion

The present study provided the first estimate of the abundance of microplastics in the Krka River estuary across the salinity gradient. This is significant because the estimate of microplastic abundance only in the SL is a significant underestimate of plastic contamination. The abundance and distribution of microplastics is determined by their properties, such as size, density, and settling velocity. It cannot be assumed that all microplastics exhibit the same behaviour in dispersion patterns. Because estuaries are mostly eutrophic environments with a vast number of fouling organisms, they can contaminate microplastic particles and alter their buoyancy properties. However, little information is available on the fragmentation dispersion and deposition in estuarine ecosystems [35]. According to [8], neutrally buoyant microplastics spread more easily throughout the water column but generally remain in the SL. Additionally, they are more likely to be flushed out of the estuary by tides than heavier ones. Heavier microplastics usually subside in the lower part of the water column and accumulate. They also tend to be finer in structure, as they are exposed to longer residence time and thus increased fragmentation. Particle size distribution is an environmentally relevant parameter to analyze because the toxicity of microplastics and ingestion of microplastics by aquatic organisms are size-dependent [36]. Density and settling velocity analysis should also be conducted to confirm this hypothesis.

In addition to the intrinsic properties of microplastics, the salt-wedge structure and dynamics of the estuary plays an important role in microplastics abundance [8]. Microplastics, like any particle in suspension, are sensitive to the complex hydrodynamics of the estuary, especially tidal currents. Normally, as the tide rises, the strong density stratification typical of the salt-wedge structure dampens the turbulent mixing event, restraining transport in suspension below the pycnocline [8]. The vertical salinity gradient influences turbulent mixing in the water column as well as bed morphology, lateral circulation, wind, earth rotation, internal waves, and sediment load [37].

This study did not consider the complex hydrodynamic influence of the estuary, which is crucial to the transport and accumulation of microplastics. This would have been beyond

the scope of this study. Therefore, the focus is shifted to the impact of the salinity gradient on the distribution of microplastics in the estuary.

To date, the vertical distribution of microplastics and the mechanisms involved in the sinking process are largely unknown [38]. According to [35], the vertical salinity structure was responsible for the distribution of microplastics in a tropical estuary in Brazil. During dry seasons, when stratification is evident, the different densities in the water masses do not allow the passing of microplastics from the upper to the lower system or vice versa. Consequently, microplastics are flushed towards the sea after increased rainfall. The density of microplastic polymers is usually a definitive factor whether the particle will sink or float, but recent studies demonstrate that they can be transported up and down the water column via vertical mixing to different layers or sediment. In addition to intrinsic properties of microplastics (i.e., particle shape, size, density), environmental factors (i.e., storms, wind-driven mixing events, resuspension, turbidity currents, biofouling) play an important role in the vertical distribution of microplastic, demonstrating accumulation to specific layers in the estuarine environment [1,39]. The halocline appears to be one distinctive feature that greatly affects the vertical distribution of plastic particles, as it forms a layer of water with varying densities and temperatures that acts as a barrier. According to our results, there was no accumulation of microplastics at the halocline as proposed by [1,5]. This can be explained in part by the co-staining of natural organic matter from a terrestrial source, which was abundantly found in SL and SBL and consequently quantified as microplastics and overestimated at the expense of other layers of the water column. Certain modifications to the protocol should be made, particularly in the removal of natural organic matter. To speed up the protocol, the use of centrifugation should be considered to assist density separation as suggested by [40].

## 5. Conclusions

In this study, a total of 910 microplastic particles were collected from 12 sampling events. The number of detected particles ranged from 389 in SL to 63 in deepest ML. The highest plastic abundance was found in SL (3.68 particles/$m^3$), followed by the SBL (0.38 particles/$m^3$), then in HL (0.19 particles/$m^3$), and the lowest in ML (0.13 particles/$m^3$). The total observed area of collected particles was 336.21 $mm^2$ overall, with the maximum of 185.23 $mm^2$ found for SBL and the minimum of 11.14 $mm^2$ for ML. Benjamini–Hochberg robust post-hoc tests revealed differences in size between the upper layers (SL, SBL) and lower layers (HL, ML). However, there was no significant difference in the upper layer between SL and SBL ($\psi = 0.036$, $p = 0.504$) and in the lower layer between HL and ML ($\psi = 0.037$, $p = 0.605$). The measured values of area indicated an (i) ellipsoidal cross-sectional shape of the particles and (ii) suggesting that we should be cautious in interpreting particle size when using area and diameter interchangeably. Finally, it was found that the majority of particles (98.4–100%) belonged to the microplastic class (diameter < 5 mm), with the small microplastic class (diameter < 1 mm) being more dominant (68.5–80.9%) than the large microplastic class (1 mm < diameter < 5 mm). To conclude, the strong salinity stratification in the Krka River estuary did not alter the usual distribution of microplastic particles in the water column and there was no significant accumulation at the halocline.

**Author Contributions:** Conceptualization, N.C. (Neven Cukrov) and M.P.; methodology, M.P. and N.C. (Nuša Cukrov); software, M.P.; validation, V.C. and M.L.; formal analysis, S.G. and M.P.; investigation, M.L. and N.C. (Neven Cukrov).; resources, N.C. (Neven Cukrov); data curation, M.P.; writing—original draft preparation, M.P.; writing—review and editing, N.C. (Neven Cukrov) and V.C.; visualization, N.C. (Neven Cukrov) and S.G.; supervision, N.C. (Neven Cukrov); project administration, N.C. (Neven Cukrov); funding acquisition, N.C. (Neven Cukrov). All authors have read and agreed to the published version of the manuscript.

**Funding:** This research was funded by Croatian Science Foundation grant number IP-2019-04-5832.

**Data Availability Statement:** Not applicable.

**Acknowledgments:** This work has been supported by Croatian Science Foundation under the project lP-2019-04-5832. Moreover, the authors would like to thank scuba diver Tomislav Bulat, from marine station Martinska, Ruđer Bošković Institute for helping in sampling.

**Conflicts of Interest:** The authors declare no conflict of interest. The funders had no role in the design of the study; in the collection, analyses, or interpretation of data; in the writing of the manuscript; or in the decision to publish the results.

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
