# Peer review of "Microplastic Distribution through the Salinity Gradient in a Stratified Estuary"

_water, doi:10.3390/w14203255_

Round 1

Reviewer 1 Report

The authors assessed the distribution of microplastic through the salinity gradient of a highly stratified estuary. I welcome the idea of specifically focusing on one area. The manuscript takes into account the characteristic parameters for the mouth of the Krka River. This approach allowed for the assessment of the microplastic distribution through the salinity gradient. The article is interesting, it will provide new information on the distribution of microplastics in the river water column.

For all these reasons, which can be summarized as: presentation of novelty / interest in the topic, sufficient description of analytical methods, in-depth discussion of the results, appropriate structure I recommend the manuscript for publication after minor correction:

In section  „Organic matter removal”

Could you mention at what pH the Fenton's reagent was added?

Author Response

  1. The authors assessed the distribution of microplastic through the salinity gradient of a highly stratified estuary. I welcome the idea of specifically focusing on one area. The manuscript takes into account the characteristic parameters for the mouth of the Krka River. This approach allowed for the assessment of the microplastic distribution through the salinity gradient. The article is interesting, it will provide new information on the distribution of microplastics in the river water column.

For all these reasons, which can be summarized as: presentation of novelty / interest in the topic, sufficient description of analytical methods, in-depth discussion of the results, appropriate structure I recommend the manuscript for publication after minor correction:

In section „Organic matter removal”

Could you mention at what pH the Fenton's reagent was added?

Response- Thank you sincerely for taking interest in this research topic and for your comments. We have added the pH of the Fenton’s reagent in the revised manuscript (pH = 2.8).

Reviewer 2 Report

The research meets the minimum specifications of the journal. I recommend posting once the general English of the document improves a bit.

Author Response

  1. The research meets the minimum specifications of the journal. I recommend posting once the general English of the document improves a bit.

Response- Thank you for taking the time to read the paper and for your comments. We tried  to improve our language in the paper so that readers are able to apprehend it quickly.

Reviewer 3 Report

Review of: “Microplastic Distribution through the Salinity Gradient in a Stratified Estuary for water (water-1942784).

Overview: This study evaluated microplastics distribution in the Krka River Estuary. We have some specifics for you.

Specific:

·        Introduction: Indent the first line of each paragraph with 2 characters.

·        Line 23: Please give the definition of microplastics.

·        Line 79-89: do you think these factors would affect the results? Why did you choose sampling time in June? Do you think your results could reflect the true contamination situation in the Krka River Estuary?

·        Line 132-136: why did you use a 250 µm sieve?

·        Line 139-144: change “ml” into “mL”, change “hour” into “h”. Check them in the whole MS.

·        Line 247-252: Please show the specific sampling sites in “2.3. Microplastics sampling”.

Author Response

  1. Introduction: Indent the first line of each paragraph with 2 characters.

Response- Thank you very much for your valuable suggestion. We have addressed it during the revision of the manuscript.

  1. Line 23: Please give the definition of microplastics.

Response- Reviewer’s suggestion is incorporated in the revised version of the manuscript.

  1. Line 79-89: do you think these factors would affect the results? Why did you choose sampling time in June? Do you think your results could reflect the true contamination situation in the Krka River Estuary?

Response- Thank you so much for your comments. These factors could probably affect the results, but we cannot conclude this from our research. We chose to sample in June because it is the last month before the tourist season starts, so that our results would not be affected by the increased anthropogenic pressure. Our results cannot reflect the true contamination situation in the Krka River Estuary, since we did not carry out a complete spatio-temporal analysis of the area.

  1. Line 132-136: why did you use a 250 µm sieve?

Response- Thank you for your comment. We used a 250 µm sieve because we thought it was the most suitable.

  1. Line 139-144: change “ml” into “mL”, change “hour” into “h”. Check them in the whole MS.

Response- Thank you so much for your minute observation. We have revised this in the whole manuscript.

  1. Line 247-252: Please show the specific sampling sites in “2.3. Microplastics sampling”.

Response- Thank you for your valuable suggestion. You can find the sampling site in Figure 1., where the sampling area is indicated with a dark ellipse. The boat towed the sampling net for 5 minutes for each sampling event in the sampling area.